# Hospital pharmacists', doctors' and nurses' perceptions of intra- and inter- professional communication in the context of electronic prescribing and medication administration systems: A qualitative study

**Soomal Mohsin-Shaikh**[1], **Ann Blandford**[2], **Bryony Dean Franklin**[1,3] *

**1** Research Department of Practice and Policy, School of Pharmacy, University College London, London, United Kingdom, **2** UCL Interaction Centre, University College London, London, United Kingdom, **3** Centre for Medication Safety and Service Quality, Pharmacy Department, Imperial College Healthcare NHS Trust, London, United Kingdom

* bryony.deanfranklin@ucl.ac.uk

**Data Availability Statement:** The datasets generated and analyzed during the current study

## Abstract

### Background

Effective, integrated and coordinated communication is essential in providing high quality patient care. Little prior research has detailed the impact of electronic prescribing and medication administration (ePMA) systems on healthcare professionals' (HCPs') communication. We investigated hospital pharmacists', doctors' and nurses' perceptions of how ePMA systems have affected, or are expected to affect, the way they communicate with each other in an inpatient setting.

### Methods

A qualitative study in three English NHS hospitals: two used different inpatient ePMA systems, and one used paper-based prescribing. We conducted focus groups with pharmacists, and semi-structured individual interviews with doctors and nurses. Transcribed data were analysed inductively using thematic analysis.

### Results

Nine focus groups, three at each hospital, were conducted with pharmacists with different levels of seniority (58 pharmacists in total). Fourteen doctors and twelve nurses took part in individual interviews. Four themes were generated: modes of communication, reduced pharmacist visibility, system limitations, and future aspirations for ePMA. Whether working with ePMA or paper-based systems, all participants preferred to communicate face-to-face to facilitate collaborative discussions regarding patient care. Participants perceived that ePMA reduced contact time with patients. Pharmacists perceived that both their physical ward presence and their written communication on medication charts had reduced since introduction of ePMA. Doctors felt they were now less likely to ask pharmacists questions

are not publicly available in their entirety due to the terms of our ethical approval. Extracts can be made available on request by emailing the ethics committee lead at UCL School of Pharmacy on sop.ea@ucl.ac.uk

**Funding:** SMS received funding via a University College London (UCL) Impact Award in partnership with Cerner Limited. BDF is also supported by the National Institute for Health Research (NIHR) North West London Patient Safety Research Collaboration and the NIHR Health Protection Research Unit in Healthcare Associated Infection and Antimicrobial Resistance at Imperial College London in partnership with Public Health England (PHE). The views expressed are those of the authors and not necessarily those of the NHS, the NIHR, PHE, the Department of Health and Social Care, or Cerner Limited. The funders did not have any role in study design, data collection and analysis, decision to publish or preparation of the manuscript.

**Competing interests:** SMS was part funded by Cerner Limited. Cerner Limited is a vendor of an electronic prescribing system. This does not alter our adherence to PLOS ONE policies on sharing data and materials.

due to pharmacists' reduced physical presence on the ward. Hardware and software limitations were identified by all HCPs, with suggestions made for future developments to better support communication.

## Conclusion

ePMA does not necessarily support communication among HCPs. Pharmacists and doctors were also concerned that ePMA reduces communication between themselves and their patients. Hospital managers should ensure sufficient hardware for HCPs, including pharmacists, to conduct their work in clinical areas, and work with ePMA system suppliers to develop ways of enhancing, rather than inhibiting, communication.

## Introduction

The introduction of inpatient electronic prescribing and medication administration (ePMA) systems has affected the ways in which healthcare professionals (HCPs) work. A recent systematic review highlights varying effects on communication, time taken to complete tasks, clinical workflow, and workarounds [1]. In some settings, the term computerised provider order entry (CPOE) is used instead of ePMA, where the scope of CPOE may also include other types of medical order such as laboratory tests and radiology [2]. The biggest information repository in health care arguably lies in the people working in it, and the biggest information system is the web of conversations that link the actions of these individuals [3]. Previous literature has also highlighted the importance of effective communication for continuity of care and patient safety [4]. The existing literature on ePMA is largely limited to the US, and to outpatient settings, certain specialities and particular workflows; there is therefore a need to further study the impact of ePMA systems on HCPs' communication practices in an inpatient setting and outside the US.

The present study investigated the impact of ePMA on HCPs' intra- and inter-professional communication, including pharmacists, in three English NHS hospital trusts (hospital organisations). Pharmacists have been identified as an under-researched group of HCPs [1], and yet play an integral role in the management of medication for hospital inpatients. Our aim was therefore to explore UK hospital pharmacists', doctors' and nurses' perceptions of how ePMA systems have affected, or are expected to affect, the way they communicate with each other in an inpatient setting.

## Methods

### Study design

We conducted an exploratory qualitative study, with a critical realist epistemology and an interpretivist methodology. We used both focus groups and semi-structured interviews to gain an insight into different HCPs' opinions and experiences with communication in the context of ePMA. We used focus groups for pharmacists to encourage discussion among staff with different experiences. Semi-structured individual interviews were used for doctors and nurses due to anticipated challenges in scheduling focus groups for these staff. The focus group and semi-structured interview topic guides were piloted before the start of the study. The COREQ (COnsolidated criteria for REporting Qualitative research) checklist (S1 File) was used in reporting [5].

## Setting

Three NHS hospital trusts in London were invited to take part (Table 1). Trust 1 and trust 2 used different inpatient ePMA systems (Cerner and JAC respectively, both of which had been in place for at least two years); trust 3 used paper-based prescribing (i.e., doctors wrote medication orders directly onto paper medication charts) at the time of the study. The Cerner ePMA system was part of a wider electronic health records system; the JAC system did not include electronic health records but was instead integrated with the pharmacy dispensing system. Both allowed prescribing and recording of medication administration and included basic decision support. Wards at each trust received a pharmacy service typical of that in UK hospitals. A pharmacist would visit their allocated ward(s) on weekdays to provide a clinical service. During their visit, the pharmacist was responsible for ascertaining patients' medication histories, medication reconciliation, reviewing medication charts, checking discharge prescriptions and ordering medications for individual inpatients and ward stock as needed. The pharmacist would also provide patient education regarding new, changed or stopped medications and respond to medication-related queries from patients and staff.

## Recruitment and consent

A senior member of the pharmacy team and/or research department at each of the participating hospital trusts was contacted and asked if they would be willing to be a local co-ordinator to recruit participants. The local co-ordinators emailed pharmacists, medical and nursing staff inviting them to take part, in August to October 2018, using a mixture of convenience and snowball sampling [6]. Potential participants were asked to email the first author directly if they were interested in taking part. Respondents were then emailed an information leaflet and invited to give written consent if willing to take part. Five to eight pharmacists were expected to attend each focus group; we aimed for three or four focus groups at each hospital trust to give participant numbers consistent with previous recommendations [7, 8]. Similarly, we aimed for five semi-structured interviews with doctors and five with nurses at each participating trust, to give 15 of each profession in total. These numbers were chosen in advance based on the time available and our experience recruiting healthcare professionals to take part in similar studies. Previous studies also suggest that thematic saturation is usually reached with about 6–12 interviews, although we do also recognise the limitations of the concept of theoretical saturation [9].

**Table 1. Summary of hospital trusts involved in this study.**

|  | Trust 1 | Trust 2 | Trust 3 |
|---|---|---|---|
| Total hospitals in trust | 5 | 1 | 3 |
| Hospitals involved in this research | 3 | 1 | 1 |
| Hospital trust status | Teaching | Teaching, district general | District general |
| Total inpatient bed capacity at hospital(s) involved in the research | 1,274 | 207 | 800 |
| Prescribing system used | Cerner ePMA | JAC ePMA | Paper-based |

ePMA: electronic prescribing and medication administration.

## Inclusion and exclusion criteria

All pharmacists, of all specialities, employed at the participating trusts were eligible to take part in the focus groups. Doctors (of all grades and specialities) and nurses (NHS pay band 5 and above) were eligible to take part in the semi-structured interviews.

Other HCPs were excluded as they less often engage with ePMA systems/paper medication charts. Medical students, student nurses, pharmacy technicians and pre-registration pharmacists were also excluded.

## Focus group and semi-structured interview procedure

Focus groups and semi-structured interviews took place during participants' lunch times and were conducted face-to-face by the first author, a female hospital pharmacist and PhD student, in a hospital meeting room. The interviewer was known to some of the pharmacy staff at Trust 1. Refreshments were provided and no other non-participants were present. All were audio recorded and later transcribed by an external transcription company; the lead researcher also made field notes.

The questions participants were asked during the focus groups and semi-structured individual interviews were based on the following:

1. Participants' demographic information;

2. Previous experience with ePMA and paper medication charts;

3. Methods currently available / used to communicate information among pharmacists, doctors and nurses;

4. Type of information exchanged among pharmacists, doctors and nurses;

5. Future/redesigning of ePMA systems to support communication; and

6. Perceived advantages and disadvantages of using ePMA systems to communicate information.

Topic guides for the focus groups and interviews were piloted before use; final versions are provided in S2 and S3 Files respectively. We did not invite member-checking of transcripts.

## Data analysis

Anonymised transcripts were read and coded inductively by the first author, using reflexive thematic analysis [10]. Thematic analysis was chosen as it is particularly appropriate when exploring subjective experiences and opinions, with a reflexive approach suited to an interactive and inductive approach to analysis [10]. Analysis was facilitated by the use of NVivo 12 Pro (version 12.2.0) software. The transcripts were initially coded line-by-line with the codes then grouped to form coherent themes relating to our study aim. Coding, and identification of themes, were iterative processes, using an Excel spreadsheet to organise the codes into themes. The first author discussed the codes and the decision to incorporate them into themes with the other authors (AB and BDF). AB and BDF oversaw each stage of the analysis and provided feedback on coding and generation of themes. Any differences were addressed through discussion to reach a deeper understanding of the data and its interpretation. Codes were further explored if they aligned with the study aim.

### Ethical approvals

The study was registered with UCL data protection team (UCL Data Protection Registration Number: Z6364106/2018/05/20 health research) and approved by UCL Research Ethics Committee (Project ID number: 11927/001). Permission was obtained from the participating hospital trusts to access their sites for data collection through Health Research Authority (HRA) approval (IRAS project ID: 247707, Protocol number: 18/0293).

## Results

### Participant demographics

Fifty-eight pharmacists took part in nine focus groups (three per hospital trust), in September and October 2018. All pharmacists at the two ePMA trusts had prior experience with paper-based systems. Focus groups lasted between 39–63 minutes.

Fourteen doctors and twelve nurses took part in individual semi-structured interviews across the three trusts between September and November 2018. The majority had experience working with both ePMA and paper-based systems. One doctor at trust 3 had experience working with paper-based systems only. Interviews with nurses and doctors lasted 12–44 minutes and 16–53 minutes respectively. No participants withdrew once they had joined the focus group or interview. Fewer doctors and nurses participated at trust 3 as staff were preparing for an external inspection. Table 2 presents a summary of all participants.

### Themes

Four major themes were generated, with associated subthemes (Table 3).

**Theme one–Modes of communication.** Regardless of whether they were from an ePMA or paper-based trust, the majority of HCPs preferred face-to-face communication for both intra- and inter-professional communication, and believed it to be most effective. Several reasons were provided for this: facilitation of two-way communication, being able to communicate information quickly and clearly, information being instantly received by others, and being able to gauge the recipient's understanding through their body language.

> *". . .you can have a kind of two-way dialogue about something in a way that you can't with written communication to the same extent, for it's much slower to do that"* (Trust 1, Doctor 2)

> *". . .face-to-face, I think, is always preferred, because when you're having a conversation it might not just be exactly one point, or you might need some further clarification. And I think sometimes there can be some misunderstanding via email, for example. . .it's not always clear over the phone, whereas you can follow up with a few questions in the same [face-to-face] conversation"* (Trust 1, Nurse 5)

> *"I think another benefit of face-to-face is you can read the body language. So yes, you know whether they're actually taking you seriously or not"* (Trust 1, Focus group 1, Pharmacist 6)

ePMA was perceived to have affected modes of communication in different ways. For example, in relation to communication among professional groups, some nurses who used ePMA felt that doctors were less likely to inform them when a new medication was prescribed or changed.

**Table 2. Summary of participants.**

| | Total | Trust 1 (Cerner) | Trust 2 (JAC) | Trust 3 (Paper-based) |
|---|---|---|---|---|
| **Gender** | | | | |
| Female | 66 | 22 | 21 | 23 |
| Male | 18 | 8 | 6 | 4 |
| **Professional Group** | | | | |
| Doctors | 14 | 5 | 5 | 4 |
| Nurses | 12 | 5 | 5 | 2 |
| Pharmacists | 58 | 20 | 17 | 21 |
| **Pay band/Seniority (%)** | | | | |
| **Pharmacists** | | | | |
| 6 (newly qualified) | 12 (21%) | 6 (30%) | 6 (35%) | 0 (0%) |
| 7 | 22 (38%) | 6 (30%) | 5 (30%) | 11 (52%) |
| 8+ (most senior) | 24 (41%) | 8 (40%) | 6 (35%) | 10 (48%) |
| **Doctors** | | | | |
| Foundation year | 4 (29%) | 0 (0%) | 3 (60%) | 1 (25%) |
| Specialist trainee | 9 (64%) | 4 (80%) | 2 (40%) | 3 (75%) |
| Consultant | 1 (7%) | 1 (20%) | 0 (0%) | 0 (0%) |
| **Nurses** | | | | |
| 5 (newly qualified) | 4 (34%) | 0 (0%) | 4 (80%) | 0 (0%) |
| 6 | 2 (16%) | 2 (40%) | 0 (0%) | 0 (0%) |
| 7 | 5 (42%) | 2 (40%) | 1 (20%) | 2 (100%) |
| 8+ (most senior) | 1 (8%) | 1 (20%) | 0 (0%) | 0 (0%) |
| **Mean years of experience (range)** | | | | |
| Pharmacists | 7.4 (1–25) | 6.7 (1–25) | 7.6 (1–20) | 8.1 (2–20) |
| Doctors | 3.5 (1–10) | 5.2 (2–10) | 2 (1–4) | 3.3 (2–4) |
| Nurses | 10.8 (1–25) | 8.8 (3–21) | 10 (1–25) | 18 (17–19) |
| **Experience with electronic prescribing and medication administration (%)** | | | | |
| Doctors | 13 (93%) | 5 (100%) | 5 (100%) | 3 (75%) |
| Nurses | 12 (100%) | 5 (100%) | 5 (100%) | 2 (100%) |
| Pharmacists | 46 (79%) | 20 (100%) | 17 (100%) | 9 (43%) |

*"I do find. . .that drugs appear on [ePMA system] and they're [nurses] not told. . .we don't know the drug is on there unless they [prescriber] tell somebody. Even though it's a good system, if you don't tell us it's on there we won't know unless we check"* (Trust 1, Nurse 5)

Conversely, some staff felt that ePMA aided communication through encouraging more complete documentation.

*"And when you stop medications, they'll ask reasons for stopping and things like that. So it gives you a lot of information. When you're doing discharge summaries, you have no idea why medications are started or stopped and you sometimes end up making assumptions. So in that sense, it's quite good"* (Trust 3, Doctor 2)

At the ePMA sites, it was suggested that having remote access to the ePMA system could allow for better communication with colleagues and other HCPs. However, doctors also highlighted that having access to the medication chart remotely could lead to reduced face-to-face time with patients.

**Table 3. Themes and subthemes generated from the focus groups and semi-structured interviews.**

| | Theme | | Subthemes |
|---|---|---|---|
| 1 | **Modes of communication** | | Within professions |
| | | | Across professions |
| | | | With patients |
| 2 | **Pharmacists' visibility** | | Physical presence on ward |
| | | | Visibility of written communication |
| 3 | **System limitations affecting communication** | | Hardware |
| | | Software | Different screen views for different healthcare professionals |
| | | | Multiple user access |
| | | | Limitations of the electronic medication chart |
| 4 | **Future aspirations** | | Messaging centre |
| | | | Alerts when changes made to medication |
| | | | Electronic medication requests |
| | | | Addressing single user access |

*". . .if you're doing it [prescribing] remotely, there's a problem of not having to go to the patient's bedside to get the drug chart. It means you don't actually look at the patient, and there's a lot of information that you can get just from seeing them at the end of the bed. Perhaps the patient has a less of an opportunity to ask questions about particular medication . . .I might be less inclined to communicate our plans to the patient, as well."* (Trust 1, Doctor 4)

Pharmacists at the ePMA trusts agreed that ePMA led to reduced contact with patients, which was also a concern for pharmacists at the paper-based trust.

*". . .Less patient facing because we're going to be stuck on a computer, trying to add our bits and pieces"* (Trust 3, Focus group 2, Pharmacist 9)

**Theme two–Pharmacists' visibility.** Both doctors and pharmacists perceived that pharmacists at the ePMA trusts now had reduced physical presence on the wards. Both professions felt that remote screening of prescribed medication, and contacting prescribers via phone, had instead increased. Participants at all three trusts were concerned about the impact of this on patient care.

*"I do notice a lot of people doing their ward from their desk, which I really dislike and I think it has been fed back on some wards that people don't even know that a pharmacist's there, or who their pharmacist is any more"* (Trust 1, Focus group 2, Pharmacist 12)

*"You don't need to physically leave your chair. That is the biggest disadvantage, because everything's available on the screen, you don't have to look at anything or anyone"* (Trust 3, Focus group 3, Pharmacist 18)

Doctors at the two ePMA trusts believed that pharmacists' written communication had also reduced since the introduction of ePMA. In particular, they highlighted that there was no longer any 'green pen', as traditionally used by UK pharmacists to annotate paper medication charts.

*"You would write on a drug chart. . .and then the next day you would go. . .and some things would be changed in green pen. And it's those little things that are lost because now you only really get told about the things that are worth saving up for the whole week or worth taking the time to call you. So, it's only the big things that you hear about. All the little things that were wrong with your prescriptions you don't hear about."* (Trust 1, Doctor 1)

**Theme three–System limitations affecting communication.** From pharmacists' perspectives, the main limitation of ePMA both experienced at the ePMA trusts and anticipated at the paper-based trust was insufficient computers in patient areas. If there were too few computers, pharmacists were unable to carry out their work on the wards and instead had to work from offices away from the ward. This change in working practice could have contributed to the lack of pharmacists' presence on the ward noted by the other HCPs as above.

*"I also think if we're using electronic prescribing systems [it's] dependent on us having access to a computer all the time, and that might not be the case all the time. If you're on the ward for example. . .you might have computers on wheels, so we don't have access straight away."* (Trust 1, Focus group 2, Pharmacist 14)

*". . .finding computers that are working and have the [ePMA system] available. Because. . . if the doctors are trying to access it, the nurses are trying to get it, the pharmacists are trying to get on it, there's going to be a lot of people trying to. . .use all of those computers"* (Trust 3, Focus group 2, Pharmacist 14)

In relation to the software, doctors and pharmacists at the two ePMA trusts expressed their frustrations regarding the different places in which information could be documented. They were concerned that staff of other professions would not be able to find the information they documented on the ePMA system.

*"I feel like you have to look at different screens to get the same picture."* (Trust 1, Focus group 3, Pharmacist 16)

Particularly at trust 1, doctors reported that the electronic medication chart often did not display all the information they needed at first glance and some were unsure of where to find additional information. Doctors also highlighted that their screen layout differed from that used by nurses, presenting a challenge to them when a nurse queried a medication order.

*"Not all of the systems show you whether your prescription has been checked or how it's been modified, as well. So, it's less easy to see at a glance, like you could with paper, how that script [medication order] has been changed to then see what you need to do differently next time."* (Trust 1, Doctor 2)

*"I also am aware that the nurses look at a different screen than what I use often. . .So, if they're ever asking me. . .a question and showing me the screen, I will ask them to flick the screen to my view so I can understand what they're talking about. So, then, we're not even singing off the same hymn sheet"* (Trust 1, Doctor 4)

**Theme four–Future aspirations.** Participants at all three trusts suggested improvements for future ePMA systems to enhance communication among HCPs. All three HCP groups

from all three trusts suggested an electronic messaging centre for non-urgent information to be communicated to other HCPs.

> *"In terms of communication though, there's no messaging system. Maybe it'd be good if the pharmacist, for example, who'd done a drug reconciliation, can write, not as a pop up, but as a message system. So, like a new system feature within [the ePMA system] to say that regular meds [medications] have been charted. And then. . .you'll see an inbox. . .That'll be good."* (Trust 2, Doctor 3)

Nurses at the two ePMA trusts stated that it would be useful and time-saving for them to be notified electronically when a new medicine had been ordered, when a medication had been altered by the prescriber or when the discharge medication was ready.

> *". . .but maybe a system where they can inform you on [the ePMA system] that it's [change to medication] been done or they can inform you, I've written up the dose, or the TTAs ['to take away' prescription] have been done, rather than you then having to chase it up constantly."* (Trust 2, Nurse 2)

Pharmacists and nurses also suggested that medication requests for patients should be made electronically rather than by nurses using faxes or writing details in a medication order book on the ward as was currently the case. It was perceived that this would improve efficiency, save time and reduce transcribing errors.

> *"Often finding a fax that works is quite hard. So, faxing things is quite dated whereas electronic would be so much better"* (Trust 1, Nurse 2)

Finally, HCPs expressed frustration with the system at trust 2, where only one user could access a given patient's record at a time, and wanted this to be addressed such as by allowing read-only access to other users while a primary user was editing that patient's record.

## Discussion

This qualitative study provides new understandings of hospital doctors', nurses' and pharmacists' perceptions and desires regarding inter- and intra-professional communication in the context of ePMA. Our findings support those of previous research demonstrating the importance of collaborative communication and of face-to-face communication specifically [11, 12]. Participants highlighted that non-verbal communication formed an important part of such communication, with body language and facial expressions known to play an important role [11]. However, our findings suggest that ePMA may lead to reduced face-to-face communication, particularly between pharmacists and doctors. This may have important patient safety implications. For example, previous research demonstrates that pharmacists' recommendations have a higher acceptance rate by prescribers when delivered verbally compared to other methods [12].

Doctors at the two trusts using ePMA also perceived a reduction in pharmacists' presence on the wards: a finding not previously documented in the literature. As a result, doctors felt they were now less likely to ask pharmacists questions as they would need to take an additional step to contact them via phone or pager. Staff were also concerned about reduced face-to-face communication affecting rapport. The importance of relationship-building and establishing rapport was also described in a previous Australian study, in which doctors desired an accessible team-based relationship with pharmacists who could provide them with continuous

support [13]. Doctors in the current study also felt that they were receiving less feedback from pharmacists as recommendations were less obvious on ePMA. Participants at the two ePMA trusts raised a further issue regarding different HCPs being presented with different information or screen layouts, such that HCPs were not 'singing off the same hymn sheet'. This may be due to different access levels for different HCP groups. However, participants highlighted that this could hinder collaborative discussions. Inconsistences in interfaces have also been highlighted as a limitation to workflow [14].

## Implications for practice

This study suggests that ePMA systems may be inhibiting inter- and intra-professional communication, and that there may be opportunities for improving how they are used to facilitate communication. HCPs highlighted concerns regarding the system's impact on their work and relationships with other HCPs. This research also adds cumulative validity to existing literature that highlights the importance of face-to-face verbal communication. HCPs should therefore continue to prioritise face-to-face communication wherever practical, to build rapport and facilitate a more team-based approach to patient care.

Information technology teams and hospital managers should review availability of terminals in clinical areas to ensure sufficient hardware for HCPs, including pharmacists, to conduct their work while maintaining a physical presence. Education and training also remain important in ensuring that all HCPs are aware of the limitations of ePMA.

ePMA system providers should elicit regular feedback from their users and incorporate this into ongoing development of their solutions, including enhancing features such as inter- and intra-professional messaging to improve communication, collaboration and workflow. The ability to add coloured text may be a simple solution in increasing the visibility of pharmacists' written communication on ePMA charts, in line with what these UK HCPs previously appreciated when using paper medication charts.

## Strengths and limitations

A strength of our study is that more than 80 HCPs across three NHS trusts provided a variety of experiences and opinions; these comprised doctors, nurses and pharmacists of different grades and specialities. The themes identified were similar across professions and trusts, suggesting that our findings are generalizable beyond these three organisations.

Limitations include focus groups and semi-structured interviews being held during participants' lunchtimes. While this facilitated recruitment, it meant that the duration was generally limited to an hour, which limited opportunity for further discussion. In addition, pharmacist focus groups included both junior and senior pharmacists which might have resulted in junior pharmacists feeling less able to express concerns. Most of the doctors who took part were less than four years qualified and so the views of more experienced doctors such as consultants may not have been captured, although junior doctors often have the most interaction with ePMA systems as they do most of the prescribing for hospital inpatients in UK hospitals [15]. The first and last authors are pharmacists, which could have introduced bias through bringing a more pharmacy-centric perspective. Participants tended to focus on the disadvantages of ePMA; it is not clear whether this reflects their genuine experience or a tendency to focus on negative consequences. Finally, the data were collected prior to the COVID-19 pandemic, which may have resulted in changes in communication practices and the use of ePMA. For example, staff may now be more likely to work remotely, introducing additional communication challenges.

## Future research

Research is now needed to develop and evaluate solutions to improve inter- and intra-professional communication in the context of ePMA, as well as studying a wider range of ePMA systems than those studied here. Subsequent research should also include patient and carer perspectives, particularly given HCPs' concerns that ePMA has led to reduced face-to-face time with patients, as well as the perspectives of more senior doctors and any impact of remote working becoming more prevalent following the COVID-19 pandemic. Observation of communication patterns in practice is needed to complement the perceptions of HCPs elucidated here.

## Conclusion

Our findings suggest that ePMA may adversely affect intra- and inter-professional communication, with reduced pharmacist presence on the wards a particular concern. Actions are now needed to encourage and facilitate face-to-face communication where appropriate, as well as developing ways in which ePMA systems can be used to support other methods of communication.

## Supporting information

**S1 File. Consolidated criteria for reporting qualitative studies (COREQ): 32-item checklist.**
(DOCX)

**S2 File. Topic guide for focus groups with pharmacists.**
(DOCX)

**S3 File. Topic guide for semi-structured interviews with medical and nursing staff.**
(DOCX)

## Acknowledgments

We thank Gangotri Darji and Adrian Hire for their help with proofreading this work.

## Author Contributions

**Conceptualization:** Soomal Mohsin-Shaikh, Ann Blandford, Bryony Dean Franklin.

**Data curation:** Soomal Mohsin-Shaikh.

**Formal analysis:** Soomal Mohsin-Shaikh, Ann Blandford, Bryony Dean Franklin.

**Funding acquisition:** Ann Blandford, Bryony Dean Franklin.

**Investigation:** Soomal Mohsin-Shaikh.

**Methodology:** Soomal Mohsin-Shaikh, Ann Blandford, Bryony Dean Franklin.

**Project administration:** Soomal Mohsin-Shaikh.

**Supervision:** Ann Blandford, Bryony Dean Franklin.

**Validation:** Ann Blandford.

**Writing – original draft:** Soomal Mohsin-Shaikh, Ann Blandford, Bryony Dean Franklin.

**Writing – review & editing:** Soomal Mohsin-Shaikh, Ann Blandford, Bryony Dean Franklin.

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
