## [Decision Letter · Decision Letter 0]

22 May 2023

PONE-D-23-05914Hospital pharmacists’, doctors’ and nurses’ perceptions of intra- and inter- professional communication in the context of electronic prescribing and medication administration systems: a qualitative studyPLOS ONE

Dear Dr. Franklin,

Thank you for submitting your manuscript to PLOS ONE. After careful consideration, we feel that it has merit but does not fully meet PLOS ONE’s publication criteria as it currently stands. Therefore, we invite you to submit a revised version of the manuscript that addresses the points raised during the review process.

In addition to addressing the comments of reviewer 1 please could you also add in some more detail surrounding your reflexive thematic analysis. For example how did you move from codes to themes, what decisions had to be made during this process with regards to developing or removing the codes that you grouped together? Also on page 8 line 169 you say that differences in interpretation were resolved through discussion. However, this approach goes against one of the principles of reflexive TA that you are not looking to find a consensus and the use of multiple researchers is more for adding deeper interpretation and richer meaning. In light of this, please rephrase or amend this sentence. 

We look forward to receiving your revised manuscript.

Kind regards,

Heather Leggett

Academic Editor

PLOS ONE

Journal Requirements:

“We have read the journal's policy and the authors of this manuscript have the following competing interests: SMS was part funded by Cerner Limited. Cerner Limited is a vendor of an electronic prescribing system.”

Reviewers' comments:

Reviewer's Responses to Questions

**Comments to the Author**

1. Is the manuscript technically sound, and do the data support the conclusions?

Reviewer #1: Yes

Reviewer #2: Yes

2. Has the statistical analysis been performed appropriately and rigorously? 

Reviewer #1: N/A

Reviewer #2: N/A

3. Have the authors made all data underlying the findings in their manuscript fully available?

Reviewer #1: Yes

Reviewer #2: Yes

4. Is the manuscript presented in an intelligible fashion and written in standard English?

Reviewer #1: Yes

Reviewer #2: Yes

5. Review Comments to the Author

Reviewer #1: This is an interesting, important and well presented piece of research throwing new light on patient safety and professional communication issues. Some issues for clarification:

1. The data was collected five years ago and may not reflect current professional or technical practice. It would be helpful to have some understanding of any major systems changes that may have come about in the interim and also to reflect on how the current situation may be further researched. The major issue of Covid in the interim may be of significance - did this have an impact on prescribing systems? It certainly will have impacted on models of professional communication...

2. For those who are not familiar with NHS systems, it would be useful to briefly indicate the main features / differences of the ePMA systems - e.g. recording only, advisory, highlight risk issues, detect errors etc.

3. An impressive number of participants were involved but there is significant variation between professions, grades and sites in terms of who took part; the absence of consultants from two hospitals is striking. This may reflect the more involved role of junior doctors but it raises issues about the involvement, understanding and experience of those who are often making the prescribing decisions rather than simply recording them. Is there a place for further research here?

4. The methodology and study guide are appropriate - however, the balance of the data seems to reflect the disadvantages or possible improvements needed for the systems. This is understandable in this research framework but there is limited information on the reported strengths / advantages of the systems - the authors might consider if such information might be explored.

5. The discussion highlights important experience among HCPs which might not otherwise have emerged in other research settings. Broader exploration of further and up-to-date research opportunities might be of value.

Reviewer #2: Dear authors,

thank you for opportunity to review this manuscript.

In general, I consider the paper to be really well written. Background and methodology are clearly explained, results well presented. I believe the manuscript meets expected standards and I recommend the article for publishing.

6. PLOS authors have the option to publish the peer review history of their article (what does this mean?). If published, this will include your full peer review and any attached files.

Reviewer #1: **Yes: **Gerard Bury

Reviewer #2: No

---

## [Decision Letter · Decision Letter 1]

8 Nov 2023

Hospital pharmacists’, doctors’ and nurses’ perceptions of intra- and inter- professional communication in the context of electronic prescribing and medication administration systems: a qualitative study

PONE-D-23-05914R1

Dear Dr. Franklin,

We’re pleased to inform you that your manuscript has been judged scientifically suitable for publication and will be formally accepted for publication once it meets all outstanding technical requirements.

Kind regards,

James Mockridge

Staff Editor

PLOS ONE

Reviewers' comments:

Reviewer's Responses to Questions

**Comments to the Author**

1. If the authors have adequately addressed your comments raised in a previous round of review and you feel that this manuscript is now acceptable for publication, you may indicate that here to bypass the “Comments to the Author” section, enter your conflict of interest statement in the “Confidential to Editor” section, and submit your "Accept" recommendation.

Reviewer #1: All comments have been addressed

Reviewer #2: All comments have been addressed

2. Is the manuscript technically sound, and do the data support the conclusions?

Reviewer #1: Yes

Reviewer #2: Yes

3. Has the statistical analysis been performed appropriately and rigorously? 

Reviewer #1: Yes

Reviewer #2: N/A

4. Have the authors made all data underlying the findings in their manuscript fully available?

Reviewer #1: Yes

Reviewer #2: Yes

5. Is the manuscript presented in an intelligible fashion and written in standard English?

Reviewer #1: Yes

Reviewer #2: Yes

6. Review Comments to the Author

Reviewer #1: (No Response)

Reviewer #2: Dear authors,

Thank you for clarification comments and adjusting your manuscript. I recommend it for publishing.

7. PLOS authors have the option to publish the peer review history of their article (what does this mean?). If published, this will include your full peer review and any attached files.

Reviewer #1: **Yes: **Gerard Bury

Reviewer #2: No

---

## [Editor Report · Acceptance letter]

22 Nov 2023

PONE-D-23-05914R1 

Hospital pharmacists’, doctors’ and nurses’ perceptions of intra- and inter- professional communication in the context of electronic prescribing and medication administration systems: a qualitative study 

Dear Dr. Franklin:

I'm pleased to inform you that your manuscript has been deemed suitable for publication in PLOS ONE. Congratulations! Your manuscript is now with our production department. 

Kind regards, 

on behalf of

Dr James Mockridge 

Staff Editor

PLOS ONE